# Linking Cancer Stem Cell Plasticity to Therapeutic Resistance-Mechanism and Novel Therapeutic Strategies in Esophageal Cancer

**DOI:** 10.3390/cells9061481

**Published:** 2020-06-17

**Authors:** Chenghui Zhou, Ningbo Fan, Fanyu Liu, Nan Fang, Patrick S. Plum, René Thieme, Ines Gockel, Sascha Gromnitza, Axel M. Hillmer, Seung-Hun Chon, Hans A. Schlösser, Christiane J. Bruns, Yue Zhao

**Affiliations:** 1Department of General, Visceral, Cancer and Transplantation Surgery, University Hospital Cologne, 50937 Cologne, Germany; chenghui.zhou@uk-koeln.de (C.Z.); ningbo.fan@uk-koeln.de (N.F.); fanyu.liu@student.uni-tuebingen.de (F.L.); patrick.plum@uk-koeln.de (P.S.P.); seung-hun.chon@uk-koeln.de (S.-H.C.); hans.schloesser@uk-koeln.de (H.A.S.); christiane.bruns@uk-koeln.de (C.J.B.); 2Interfaculty Institute for Cell Biology, University of Tübingen, Auf der Morgenstelle 15, 72076 Tübingen, Germany; 3Singleron Biotechnologies, Yaogu Avenue 11, Nanjing 210000, China; nan@singleron.com; 4Institute of Pathology, Faculty of Medicine and University Hospital Cologne, University of Cologne, 50937 Cologne, Germany; Sascha.gromnitza@uk-koeln.de (S.G.); ahillmer@uni-koeln.de (A.M.H.); 5Department of Visceral, Transplant, Thoracic and Vascular Surgery, University Hospital of Leipzig, 4107 Leipzig, Germany; Rene.Thieme@medizin.uni-leipzig.de (R.T.); ines.gockel@medizin.uni-leipzig.de (I.G.); 6Center for Molecular Medicine Cologne, University of Cologne, 50937 Cologne, Germany

**Keywords:** esophageal cancer, heterogeneity, cancer stem cell, plasticity, therapeutic resistance

## Abstract

Esophageal cancer (EC) is an aggressive form of cancer, including squamous cell carcinoma (ESCC) and adenocarcinoma (EAC) as two predominant histological subtypes. Accumulating evidence supports the existence of cancer stem cells (CSCs) able to initiate and maintain EAC or ESCC. In this review, we aim to collect the current evidence on CSCs in esophageal cancer, including the biomarkers/characterization strategies of CSCs, heterogeneity of CSCs, and the key signaling pathways (Wnt/β-catenin, Notch, Hedgehog, YAP, JAK/STAT3) in modulating CSCs during esophageal cancer progression. Exploring the molecular mechanisms of therapy resistance in EC highlights DNA damage response (DDR), metabolic reprogramming, epithelial mesenchymal transition (EMT), and the role of the crosstalk of CSCs and their niche in the tumor progression. According to these molecular findings, potential therapeutic implications of targeting esophageal CSCs may provide novel strategies for the clinical management of esophageal cancer.

## 1. Introduction

Esophageal cancer (EC) is the 7th most commonly diagnosed cancer and the 6th leading cause of cancer-related death worldwide, with an estimated 572,000 new cases and 509,000 deaths in 2018 [1]. Esophageal adenocarcinoma (EAC) and esophageal squamous cell carcinoma (ESCC) are the two main histopathological subtypes of EC. EAC and ESCC vary in etiology and pathogenesis, genomic characteristics, geographical distribution, ethnic characteristics, and therapeutic sensitivity [2]. In addition to the common risk factors such as older age, male gender, tobacco smoking, and lower socioeconomic status, EAC is reported to be more related to obesity, gastroesophageal reflux disease (GERD), and Barrett’s esophagus, whereas ESCC is more associated to alcohol or hot beverages consumption and family history of cancer [3]. EAC exhibits frequent genomic amplifications of VEGFA, ERBB2, GATA4, GATA6, and CCNE1 as well as deletions of SMAD4, while ESCC generally presents amplifications of CCND1, SOX2, TERT, FGFR1, MDM2, NKX2-1, and/or TP63 as well as deletions of RB1 [4]. At the level of point mutations shows EAC frequent mutations in TP53, CDKN2A, ARID1A, and SMAD4 while ESCC is frequently mutated in TP53, CSMD3, NOTCH1, and PIK3CA [5,6]. EAC is more frequent in many western countries including Germany, while ESCC is the major histological type in eastern countries especially in China and Japan [7,8]. Years of efforts have improved the 5-year survival of EC from less than 5% in the 1960s to about 20% in recent decades [2]. Gradual improvement of multi-disciplinary management strategies of EC contributed to the improved therapeutic effect over time [9]. However, due to the lack of obvious symptoms at the early stage of the disease, EC patients usually have developed regional or distant metastasis at the time of diagnosis, which makes EC still a major global health care challenge. In addition, not all patients benefit from the multimodal therapies including neoadjuvant chemotherapy or perioperative chemoradiation and show no tumor response at all [10,11]. So far, the exact mechanisms underlying therapeutic resistance are often unclear.

Cancer stem cells (CSCs) are a small group of cancer cells with specific properties such as self-renewal, differentiation potential, proliferation, heterogeneity, and therapeutic resistance [12]. Since the first identification of CSC in acute myeloid leukemia (AML) by Bonnet et al. in 1990s [13], this particular subset of cells was reported in many solid tumors including gastrointestinal carcinoma [14,15]. The classic hierarchic CSC theory is that only CSCs have self-renewal ability and are able to differentiate into progenitor cells that lead to differentiated tumor cells. However, recent studies have shown the plasticity of CSCs while non-CSCs are capable of gaining stemness due to the changes in tumor microenvironment (TME) or the stimulations by cytotoxic treatments [16,17]. It is suggested that CSCs may be responsible for therapeutic resistance and are the major cellular source for tumor recurrence [12,17,18]. According to the CSCs theory, traditional cytotoxic treatments like chemotherapy and radiotherapy could eliminate rapidly proliferating non-CSC cells rather than the relatively quiescent CSCs and may stimulate non-CSCs to undergo stem-phenotypic transitions [16,17,18]. For EC patients, no significant survival benefit of an adjuvant chemotherapy or radiotherapy has been shown [19,20,21]. It has been reported that nearly 70% of patients showed limited or no response to current neoadjuvant chemotherapy and still 30–40% of patients did not achieve a satisfactory response after neoadjuvant chemoradiotherapy [10,22,23]. Moreover, long-term follow-up revealed that about 40–50% of patients developed local or distant recurrence even after radical multidisciplinary treatment [24,25,26]. In light of this relatively poor susceptibility of EC to chemo- or radiotherapy, it appears highly promising to understand the role of CSCs in EC and to explore therapeutic strategies aiming to eradicate CSCs.

In this review, we focus on the latest research findings on CSCs in EC from PubMed based on the medical subject headings of ‘esophageal cancer’, ‘esophageal adenocarcinoma’ or ‘esophageal squamous cell carcinoma’, ‘cancer stem cell’, ‘heterogeneity’ or ‘single cell’, ‘signaling pathways’, ‘chemotherapy’, ‘radiotherapy’ or ‘therapeutic resistance’, ‘prognosis’ or ‘survival’. Only peer reviewed articles written in English were included. We thoroughly discuss isolation of CSCs, their biological characteristics, TME crosstalk, therapeutic resistance, and potential novel perspectives of CSCs eradication in EC.

## 2. Isolation of Esophageal Cancer Stem Cells

The introduction of the “tumor stem cell” concept bears interesting opportunities to explore the pathogenesis of malignant tumors. Reliable and robust protocols for esophageal cancer stem cells (ECSCs) isolation and enrichment is a pivotal task to harmonize ECSC studies. In recent years, many experts have explored several separation methods of ECSCs, mainly belonging to the following two types of strategies: ECSC biomarker based and ECSC biomarker-free based.

### 2.1. ECSC Isolation—Biomarker Based

Using the specific surface or intracellular markers detectable by fluorescence-activated cell sorting (FACS) or antibodies conjugated to magnetic beads for screening of CSC is considered as one of the most authoritative methods (Figure 1).

CD44 and CD133 are multifunctional cell surface antigens that have a role in tumor proliferation, migration, invasion, and angiogenesis in several aspects of cancer cell phenotypes and have been extensively studied as single and combined CSC markers [27,28]. Single marker CD44 is suggested to be a prognostic marker for EAC and ESCC [29,30]. In particular, some studies have proposed CD44 as a CSC marker in ESCC [29,31]. Functional characteristics were also found for CD133 [32,33,34], CD271(p75NTR) [35,36,37], LgR5 [38,39,40], CD90 [41,42], ALDH1 [43,44,45], ABCG2 [33,46,47], ICAM-1[48] and ITGA7 [49]. Besides, CD44 and CD133 can be used in combination with CD24 (CD44^+^/CD24^−^) [50], CD133 (CD44^+^/CD133^+^)[51] and CXCR4 (CD133^+^/CXCR4^+^) [52] to identify esophageal CSCs (Table 1).

Other read-outs for cancer stemness associated genes such as BMI-1 [61], Nanog [58,59,60], Sox2 [53,54,55,56,57], Oct-4 [54,62,65,66], SALL4 [55,73], GLI-1 [70,71,72], Ep-CAM [67,68,69], and Podoplanin (PDPN) [74,75,76,77] are involved in regulating CSC populations, leading to enhanced proliferation, invasiveness, therapy resistance, and metastatic capacity. They could potentially act as prognostic CSC markers in esophageal cancer.

### 2.2. ECSC Isolation—Biomarker-Free

In addition to the strategies mentioned above, there are several common methods for CSC isolation independent of specific markers. Firstly, side population (SP) cells are a subpopulation of cells that can exclude dyes such as Hoechst 33342 and therefore can be identified through FACS analysis. SP cells appear to be enriched with stem cells and share many biological characteristics with both normal and cancer stem cells [78], thus, they were regarded as stem cell-like cells in numerous types of cancers including leukemia [79], multiple myeloma [80], breast cancer [81]. Several studies have isolated stem cell-like subpopulations from esophageal cancer cells using side population strategy. For example, Huang et al. [62] isolated and identified SP cells in human esophageal cancer cell lines, the cells with the strongest dye efflux activity (SP cells) in EC9706 had higher clone formation efficiency than non-SP cells. Zhang et al. [82] demonstrated that radioresistant cell lines contained higher fractions of SP cells than parent cell lines of EC. In addition, Zhang et al. [83] also reported increased SP cells in 3D tumor spheres as compared to the 2D adherent cultured cells. Our previous study [84] detected SP cells using Hoechst 33342 staining in five different esophageal cancer cell lines and provided evidence that (1) the proportion of SP cells was variable in esophageal cancer cell lines, (2) SP cells exhibited stem cell properties and were associated with chemotherapy resistance, and (3) long-term exposure to chemotherapy drugs could enrich SP cells with EMT characteristics, which might be a source for recurrence and distant metastases. Secondly, serum-free suspension culture is widely accepted as an effective method for enrichment of CSCs. Many cancer types develop microsphere cells after serum-free suspension culture and exhibit stem cell-like characteristics [83,85,86,87]. The EAC cell line OE19 built tumor spheres, when cultured in serum-free medium, with increased expression levels of CD44 and they were more resistant to radiotherapy as the parent OE19 clone [88]. Consistently, in ESCC cell lines, sphere cells isolated through the same method showed higher radio-resistance than their parental cells [89]. Spheres from the ESCC line ECA109, cultured in serum-free medium, exhibited higher proliferation rates and tumorigenicity in vivo [90]. However, the well-accepted sphere formation assay may not be always the most effective method for CSC enrichment, and the same is true for analyses of acquired therapy resistance [91]. Thirdly, radiation resistance has been identified as a major characteristic of CSCs in vitro [92,93] and accumulating evidence indicates that CSCs are mediating resistance to radiation therapy in cancer patients [94]. Thus, radio-resistance can be used to isolate CSCs. Using an in vitro isogenic model of radioresistant EAC, Lynam-Lennon et al. [95] demonstrated that radioresistant EAC cells have enhanced tumorigenicity in vivo and increased expression of CSC-associated markers as well as enhanced holo-clone forming ability. In two other studies, fractionated irradiation was applied to acquire radio-resistant esophageal cancer cells [82,96]. Both studies demonstrated that radioresistant EAC and ESCC cells showed stem cell properties both in vitro and in vivo. Lastly, attached-cell Aldefluor method (ACAM) is used to identify stem-like cells in ESCC cell lines (KY-5, KY-10, TE-1, TE-8, YES-1, YES-2), where ACAM positive cells showed significantly higher ALDH activity and higher CD44 expression than the parental cells, which may represent a strategy to identify ECSCs [97].

Some studies defined tumor transplantation assays through serial tumor transplantation in the animal models as standard to characterize CSC subpopulations [27,98]. Limiting dilution analysis of tumor transplantation assays demonstrated that ESCC cells with higher CD44 expression showed a shorter latency for visible tumor initiation after subcutaneous tumor injections into NOD/SCID mice with low doses [29], which was especially observed in EC cells with higher expression of ALDH1 and ITGA7 [44,49]. Similarly, subcutaneous injection of OE19 SP cells as well as tumor spheres generated from Eca109 cells to nude mice showed higher tumorigenicity than their parental cells [84,90]. In addition to transplantation assays, lineage tracing is a powerful technique that allows researchers to follow the fate of individual cells and their progeny and was applied as an effective method to study stem cells [99]. Using genetic in vivo lineage tracing, Mariko et al. showed that LGR5+ tumor cells have self-renewal and differentiation capacity and functionally behave as CSCs in colon cancer [100]. As to esophagus, Jiang M et al. found p63+KRT5+KRT7+ basal cells in the upper gastrointestinal tract of mice serve as a source of progenitors for the transitional epithelium that can reproduce Barrett’s metaplasia [101]. Giroux et al. found that a long-lived progenitor cell population with expression of Krt15 is able to self-renew, proliferate, and generate differentiated cells murine esophageal epithelium [102]. Although there is still limited consensus on the identification of ECSCs, increasing amounts of studies are trying to focus on the ECSCs for both pre-clinical and clinical applications. Certainly, further investigations are still necessary to find more valid, reliable, and robust methods to identify CSCs in esophageal cancer.

### 2.3. Heterogeneity and Single-Cell Analysis of ECSCs

Tumors consist of genetically and epigenetically various cell subpopulations, which is referred to as intratumor heterogeneity. The tumor clones are not equally sensitive to current treatments and are considered a major reason for cancer treatment failure [103,104]. The CSC model is one of the most popular theories to explain intratumor heterogeneity [104]. Recent development of single cell analysis and next generation sequencing technologies allows dissection of intratumor genetic and epigenetic heterogeneity at single-cell resolution, providing new insights into the roles of CSCs in tumor initiation and intratumor heterogeneity [105].

Single-cell RNA sequencing (scRNA-seq) of primary ESCC and EAC tissues successfully distinguished tumor cells from non-tumor cells and showed intrinsic molecular heterogeneity of EAC and ESCC tumors [106]. Bulk RNA-seq and scRNA-seq of paclitaxel-resistant cells and parental cells revealed that molecular mechanisms of intrinsic paclitaxel resistance were distinct from those of acquired resistance at single-cell level. This may open new options to target paclitaxel resistance in ESCC [107]. The same methodology was also applied to analyze transcriptomic dynamics of ESCC cells with acquired radio-resistance throughout exposition of ESCC cells to different doses of irradiations in vitro. The results showed that a cellular heterogeneity with distinct subpopulations existed in irradiated ESCC cells, and dynamic gene regulations were found during the acquisition of radiation resistance [108,109]. Besides, a comparison of the transcriptomic profiles of EAC and ESCC cells with high and low stemness at single-cell level revealed a stemness-associated gene expression signature in ESCC and EAC cells. EAC CSCs highly expressed cell cycle-associated genes, while genes with regard to DNA replication and DNA damage repair were mainly increased in ESCC CSCs [110]. It was reported that beside intratumor heterogeneity, an intra-CSC heterogeneity was found in hepatocellular carcinoma, where different CSC subpopulations presented phenotypes, functions, and transcriptomic heterogeneity at a single-cell level [111]. In addition to single-cell transcriptomic analysis, single-molecule epigenomic technologies now provide an opportunity to study epigenetic regulations and dynamics such as DNA methylation, chromatin accessibility, and histone modifications at unprecedented resolution [112]. Moreover, although high-throughput single-cell methods have not yet arrived in proteomics, proteomics researchers hold an optimistic view that new technologies and strategies will soon be established to successfully tally proteins at single-cell level [113]. Therefore, given the encouraging perspective of single-cell analyses in cancer research, further studies focusing on intra-CSC heterogeneity in EC cells using single-cell analyses are needed and may provide new insights in targeting ECSCs.

## 3. ECSC Signaling Pathways

A regulatory network consisting of Wnt/β-catenin, transforming growth factor-β (TGF-β)/Smad, Notch, Hedgehog, Hippo, JAK/STAT3, and PI3K/AKT/c-MYC signaling pathways controls CSC properties [114,115,116,117,118,119,120]. These important signaling pathways regulate self-renewal, proliferation, and differentiation capacity of cancer stem cells. Dysregulation of these pathways may also contribute to the undesirable progression of esophageal cancer.

Overexpression of WNT10A plays an important role in ESCC through activation of the Wnt/β-catenin signaling pathway, inducing an increase of the CD44^+^/CD24^−^ population, which can promote ESCC migration and invasion [121]. In addition, hypoxia-inducible-factor 1α (HIF-1α) has been revealed to be essential for regulating the stemness of ESCC by activating the Wnt/β-catenin pathway. Stable knockdown of HIF-1α in ESCC cells inhibited proliferation, migration, and tumor growth in vivo [122]. MicroRNA-455-3p was reported to play key roles in promoting chemoresistance in vitro and tumorigenesis of ESCC cells in vivo. Treatment with a miR-455-3p antagomir could sensitize ESCC cells to cisplatin and reduce the subpopulations of CD90^+^ and CD271^+^ (tumor-initiating cells) T-ICs via inactivation of Wnt/β-catenin and TGF-β signaling pathways [123]. Interestingly, SB525334, a TGF-β1 inhibitor, can significantly inhibit the migration and invasion of sphere-forming stem-like cells of KYSE70 and TE1, which display an increased self-renewal capacity, chemoresistance in vitro, and tumorigenesis in vivo [124].

The Notch signaling pathway plays an important role in regulating cell differentiation and proliferation during embryogenesis and normal tissue homeostasis, which have also been implicated in tumorigenesis including development of esophageal cancer [125]. Mastermind like1 (MAML1) is a key transcription coactivator of this pathway, which could promote the aggressiveness of ESCC through an upregulation of the EMT marker TWIST1 and increase the therapy resistance of ESCC stem cells, respectively [126,127]. In addition, Notch signaling is frequently activated in poorly differentiated tumors and drives a CSC phenotype. By using patient derived xenograft models and primary cell lines, several studies have demonstrated that Notch signaling is critical for CSC capacity and able to drive stemness and tumorigenicity of EAC [128].

Glioma-associated oncogene homolog 1 (Gli-1) is a key mediator of the Hedgehog (Hh) pathway. As a transcription factor of the Hh pathway, Gli-1 mediates therapy resistance in a study with 5-FU or radiation resistant EAC cell lines (SKGT4 (SK4) and Flo-1) [70]. And Gli-1 nuclear expression was identified as a strong and independent predictor of poor response to chemoradiation, early relapse and poor prognosis in ESCC after chemoradiotherapy (CRT) [70,129]. Gli-1 expression was observed in 28.3% of ESCC and showed strong correlation with the stemness genes SOX9 and CD44, which were associated to poor prognosis in ESCC patients [72]. Furthermore, Isohata et al. reported an existing crosstalk between Hh pathways and EMT pathways in ESCC since EMT regulator SIP1 is a downstream target of Gli-1 [130], indicating potential Hh pathway regulation on EMT state of ESCC. Furthermore, Patched1 (PTCH1), another key mediator of the Hedgehog (Hh) pathway, together with Sonic Hedgehog (SHH), one of mammalian HH ligands, were significantly enriched in EC resection tissue from the patients with minimal-residual disease (MRD) after receiving neoadjuvant chemoradiation (nCRT), and PTCH1 is upregulated in CD44^+^/CD24^−^ CSC population in both EAC (OE33) and ESCC (OE21) cell lines [131]. This study demonstrated that the HH pathway might regulate CD44^+^/CD24^−^ CSC populations and increase the cancer stemness and therapy resistance.

The Hippo pathway and its downstream effector Yes-associated protein (YAP) have been proposed to be regulators of organ size, cell proliferation, and stem cell properties in a variety of cancers [132,133,134,135]. Recent studies have demonstrated that genetic or pharmacological inhibition of YAP could repress CSC-like properties in vitro and attenuate tumor growth and CSC marker expression in ESCC xenograft models by directly activating its downstream target SOX9 [136]. Consistently, another study has shown that YAP1 driven SOX9 expression was a major determinant of CSC properties in both ESCC and EAC [137]. Moreover, YAP1 could confer therapy resistance and increase cell proliferation in EC cells by upregulating epidermal growth factor receptor (EGFR) [138].

The JAK/STAT3 signaling pathway plays a prominent role in mediating tumor cell proliferation, survival, invasion, and metastasis in different types of cancer [117]. Genistein, an angiogenesis inhibitor belonging to the category of isoflavones, suppressed the JAK1/2-STAT3 pathway by decreasing EGFR expression, significantly inhibiting esophageal cell proliferation in vitro, and tumorigenesis in vivo [139]. In addition, further research demonstrated that the suppression of the JAK/STAT3 pathway could inhibit ESCC cell proliferation in vitro [140,141].

Additionally, the PI3K/AKT/c-MYC signaling axis promotes cancer stem-like feature acquisition in ESCC cell lines [120]. The study found that the cell subset responsive to a Sox2 regulatory region (SRR2) reporter (RR cells) isolated from these ESCC cell lines contained significantly higher proportions of CD44-high and ALDH1A1-high cells. The authors demonstrated that the PI3K-AKT pathway regulates the RR phenotype and promotes its CSC-like features by upregulating c-MYC [120]. It should be noted that PI3K activation was observed to be related to human papillomavirus (HPV) oncogene repression in HPV-positive cervical cancer cells, contributing to therapy resistance, immune evasion, and tumor recurrence [142]. Given the clinical and experimental evidence showing a cross-talk between HPV infection status and CSC functions in oropharyngeal cancer as well as head and neck carcinomas, virus infection and related inflammation response may as well participate in the regulation of CSCs [143,144]. However, there is still limited evidence linking HPV or other viral infections to CSC in esophageal cancer. This aspect may deserve further investigation.

## 4. Therapeutic Resistance and CSC in EC

Primary and secondary resistance are major obstacles of conventional therapeutic strategies for esophageal cancer. CSCs are frequently resistant to established therapies and may be the primary cellular source underlying resistance. Several potential mechanisms mediating CSCs-induced therapeutic resistance have been reported, including relative quiescent status with enhanced DNA repair capacity, elevated drug export efficiency, improved protection against reactive oxygen species (ROS), and the protective CSC niche in the TME [145,146,147,148].

It is well-established that normal stem cells (SCs) present a reversible quiescent state that is insensitive to cytotoxic treatments, which mainly interfere with the mitotic system of proliferating cells [149,150,151]. Adult stem cells have a very robust DNA damage response (DDR) system to maintain genomic integrity and protect cellular homeostasis in response to stress [152,153]. These features also exist in CSCs [154,155]. In EC, CSCs isolated from the ESCC cell line EC9706 were resistant to DNA damage through impaired induction of p53 and declined G1 checkpoint arrest, and presented a slow-cycling status with a lower level of phosphorylated Stat3, c-Myc, and a higher level of p27 when compared with the non-CSCs [156]. Single cell analysis revealed that overexpression of cell cycle-associated genes, DNA replication modulating genes, as well as DNA damage repair regulating genes were significantly correlated with stem cell-like properties in both EAC and ESCC [110]. Besides, CD133^+^ ESCC cells are strongly resistant to conventional cytotoxic drugs [34]. It was reported that CD133^+^ glioma stem cells are resistant to radiotherapy through the activation of the DNA damage checkpoint with enhanced DNA damage repair response [157]. A similar mechanism may mediate therapy resistance of CD133^+^ cells in ESCC as well.

Multidrug resistance (MDR) describes the phenomenon that cancer cells can be cross-resistant to several structurally and functionally different drugs [158]. A major mechanism of MDR is an altered cell membrane transport system that can pump cytotoxic drugs out of the cancer cells before irreversible DNA damage occurs. For instance, overexpression of the ATP-binding cassette (ABC) transporter family is well established in drug resistant cells [159,160]. ABCB1 and ABCG2 are widely-studied members of the ABC family and are related to CSC induced chemoresistance in many solid tumors such as breast, colon, and lung cancer [161,162]. ABCB1 and ABCG2 were also found to be remarkably upregulated in ESCC CSCs with an enhanced resistance to cisplatin as compared to non-CSCs [83]. High expression of ABCG2 correlates with poor survival in ESCC patients [33,47]. ESCC cells with ABCG2 overexpression showed cross-resistance to both irinotecan and 5-FU through the activation of the AhR pathways, which could be reversed by targeting AhR to further inhibit ABCG2 expression [163,164]. To reverse MDR in cancer cells, modulating ROS may be a viable strategy. It has been suggested that MDR cancer cells are more susceptible to alterations in ROS levels and can be sensitized to cytotoxic drugs after improving the ROS level by targeting relative modulators [165,166]. However, ROS levels were found to be decreased in many types of CSCs [167]. Among the potential mechanisms, the aldehyde dehydrogenases (ALDHs) play an important role in reducing ROS level within CSCs. ALDHs are a group of enzymes that catalyze the oxidation of aldehydes into less toxic carboxylic acids, which are commonly regarded as detoxifying enzymes [168]. As the best-studied ALDH isoform, ALDH1 was reported to decrease ROS levels through the activation of antioxidant systems [169]. Detecting of ALDH1 activity was also widely used as a classic assay to identify CSCs in a variety of cancers including EC [170]. ALDH1^+^ cells in ESCC present typical stem cell-like properties as well as higher invasive and metastatic capabilities as compared to ALDH1^−^ cells [44,45]. Clinical data suggested that EC patients with high expression of ALDH1 were more resistant to clinical interventions and had a poor long-term prognosis [44]. Taken together, the MDR of ECSCs may be attributed to the enhanced membrane pump-out ability and the decreased ROC level within tumor cells.

Normal stem cells reside in a “stem cell niche”, which refers to a dynamic microenvironment that balances the stem cell activity to govern tissue homeostasis under diverse conditions [171]. A similar concept “cancer stem cell niche” states that CSCs might localize in a protective niche within the TME, which is critical for maintaining the biological function of CSC [145,172]. As a major component of the TME, cancer-associated fibroblasts (CAFs) play a pivotal role in forming the CSC niche, promoting tumorigenesis and inducing therapeutic resistance [173]. ESCC cells co-cultured with CAFs showed significantly altered gene expression in the TME including matricellular proteins, growth factors, cytokines, chemokines, EMT-related genes, and components of inflammatory signaling pathways [174]. The cross-talk between EC cells and CAFs might be mediated by IL-6 through STAT3 and ERK1/2 signaling pathways and showed suppressed tumorigenesis both, in ESCC and EAC [174]. CAFs in ESCC were also reported to cause radio-resistance by regulating DNA damage response though promoting long noncoding RNA (lncRNA) DNM3OS expression via PDGFβ/PDGFRβ/FOXO1 signaling pathway [175]. EMT is another key process that may interact with CSC plasticity and TME. Studies have proved that stromal constituents of the TME can activate EMT through secreting various chemokines, cytokines, and activating several signaling pathways such as TGFβ, WNTs, NOTCH, and Hedgehog to maintain cancer stemness and promote tumor progression and metastasis in several types of cancers [176], including EC [177]. EC cells that underwent EMT presented an enhanced radiation resistance with improved DNA repair ability [178,179]. CSCs may also in turn modify their niche through activating EMT to reform adjacent stromal cells into a relatively undifferentiated status, which then reinforce the CSC plasticity as well as maintain the protective niche [147]. Some studies have reported that CSCs could usually be located in a hypoxic region in the TME [180], hypoxic condition of the CSC niche can also induce EMT as well as decrease inner ROS levels, which can further maintain cancer stemness and contribute to therapeutic resistance [181,182].

Up to date, many efforts are dedicated to exploring the underlying mechanisms of CSCs-induced therapeutic resistance and provide insights in cancer treatment. Among them, targeting CSCs to reverse treatment failure is one of the most promising strategies that may bring long-term benefits to cancer patients.

## 5. Therapeutic Strategies Targeting CSC in EC

The unsatisfying results of conventional chemo- and radio-therapeutic strategies highlight the clinical need of more effective therapeutic compounds, especially those which could prevent tumor recurrence. Among all the potential targets, CSCs together with the niche/TME have attracted attention as targets for pre-clinical and clinical approaches.

As discussed above, CSCs carrying variable surface markers allow the differentiation of highly malignant CSCs from normal cancer cells, many strategies based on surficial molecules and downstream compounds have also been developed to optimize treatment response. For instance, in 2016 a phase I study targeting CD44, a cancer cell progenitor marker of EAC, achieved a limited clinical benefit among AML patients. However, it suggested a promising combination therapy with other cytotoxic agents [183]. BMS-833923 is a potent and specific inhibitor of SMO in the Hh pathway [184], which is currently tested in a phase I trial (NCT00909402: completed, but results are not published yet) evaluating inhibition of SMO as a first-line therapy for unresectable metastatic EC patients in combination with Cisplatin and Capecitabine. Additionally, Taladegib (LY-2940680) an alternative small molecule interfering with the Hh cascade through binding to the SMO receptor is currently evaluated (NCT02530437: active, not recruiting) [185]. The amplification or drug induced overexpression of EGFR has long been considered as a marker for resistance and tumor progression [186]. Targeted therapy based on anti-EGFR monoclonal antibodies (mAb) e.g., nimotuzumab, plus irinotecan, a common medication for gastric cancer treatment, showed potential improvement in EGFR positive patients of advanced gastric cancer [187].

It is now widely accepted that the oncogenesis and tumor heterogeneity are not exclusively dependent on aberrations or mutations of tumor cells, but also accompanied by the dynamic changes of microenvironmental compositions as well as the state and properties of surrounding stromal cells [188,189]. Apart from malignant tumor cells, the highly diverse cell types in the TME, mainly including CAFs, immune cells, vascular endothelial cells, and mesenchymal stem cells [190,191,192], are genetically stable, thus could be utilized as multiple targets in cancer therapy. A series of efforts has been undertaken to investigate the prominent role of the CXCL12/CXCR4 axis in cancer progression of different cancer entities including both histologic subtypes of EC [193,194,195]. Combination therapies with the CXCR4 antagonist AMD3100 counteracted the resistance of pancreatic cancer cells to gemcitabine via Fak, ERK, and Akt pathways [196]. Furthermore, co-administration of AMD3100 and anti-PD-L1 antibody diminished stroma–cancer cell interaction together with an improved immune response [197]. Gockel et al. validated CXCR4 expression in 94.1% of resectable ESCC patients (54.9% weak expression vs. 45.1% strong expression) and 89.1% resectable EAC patients (71.7% weak expression vs. 29.3% strong expression), strong CXCR4 expression was supposed to be relevant to poor prognosis in both subtypes [198]. A reduced cell proliferation rate was observed in the EAC cell line OE19 treated with AMD3100, smaller primary tumor size, and fewer metastatic spread to lung, liver, and lymph node were further confirmed in OE19 injected mice under AMD3100 treatment [199]. Another approach targeting the TME arose from the observation of overexpressed VEGF/VEGF receptor system which initiates pathological angiogenesis in tumor niches. By blocking VEGFR-2 with ramucirumab, Fuchs et al. validated a significantly prolonged overall survival in a phase 3 trial among advanced gastric or gastro-esophageal junction adenocarcinoma (REGARD) patients [200]. Since rising evidence over past years supports the hypothesis that stem cells in gastroesophageal junction (GEJ) invoke very likely the occurrence of Barrett’s esophagus and EAC [201], antagonising VEGFR-2 could also be feasible in EAC.

Over the past decades, immunotherapy has been confirmed as an additional approach able to control different types of cancer and sometimes effective in patients resistant to conventional therapies [185]. Immune checkpoint receptors such as PD-1 and CTLA-4 are favorable drug targets as they release suppressive signals upon the activation by tumor cells. Up to 40% of GEJ adenocarcinoma cases express PD-L1 [202,203]. Blockade of PD-1 with the mAb Nivolumab revealed encouraging results with a survival benefit in advanced gastric or gastro-esophageal junction adenocarcinoma patients’ refractory to at least two chemotherapy regimens (ONO-4538-12, ATTRACTION-2) in a recent phase III trial in Asian patients [204]. Other novel immunological strategies involving genetically engineered immune cells (e.g., CAR T) could also elevate anti-CSC efficiency. Genetic knockdown of PD-1 alone, or in combination with chimeric antigen receptor (CAR) T cells, which carries a predefined affinity to a given tumor antigen, could significantly enhance immune reaction against a desired tumor type. Shannon et al. reported the persistent ability of CAR T cell CTL019 in monitoring ALL relapse [205]. However, in EAC patients one related clinical trial is still ongoing (NCT03706326: recruiting) and the result of another phase I study remains to be published (NCT03081715: completed). Similar to T cells, cytotoxic NK cells of the intrinsic immune system can induce apoptosis of viral infected or transformed autologous cells e.g., cancer cells via pore-forming and release of granzyme B [206]. CSCs are confirmed to be a small subset of cells with sparse MHC class I molecules (MHC-I), thus insinuates their blunt reaction to CD8^+^ T cells [207], which in term fails to silence NK cells as NK cell tolerance to self-tissue is maintained in the presence of MHC-I [208]. Additional studies have reported susceptibility of CSCs to allogeneic NK cells in diverse solid tumor types in pre-clinical models, such as colorectal cancer [209] and glioblastoma [210]. In another pre-clinical model, increased expression of the NKG2D ligands ULBP1, ULBP2, and MICA sensitized CD44^+^/CD24− breast CSCs to killing by previously activated NK cells by IL-2 and IL-15 [211]. Such investigations have ignited the passion of EC researchers to transform it into the clinical field (NCT02843581: completed) and the result is awaited by the community. The susceptibility of CSCs to cancer immunotherapy is poorly investigated and may represent an important mechanism underlying long-term benefit from these novel therapies.

According to the tumor cell subclone functional diversity model proposed by Kreso et al., conventional radio- and chemotherapy targeting dividing cancer cells collapsed to diminish dormant (stem cell-like) cells within tumor entities could account for the post-treatment repopulation of tumor cells [212]. The same group also successfully observed that the previously dormant cell lineage—a minor bunch among tumor initiating cells (T-ICs), survived from chemotherapy and contributed to tumor regrowth in colorectal cancer [213]. In glioblastoma, tumor regrowth was significantly halted after the ablation of a subset of stem cell like endogenous tumor cells [214]. Both studies favorably provided direct evidence of the tumor reproducing capacity of T-ICs. Thus, it is feasible to hitch CSCs to EC recurrence albeit similar investigations are still scarce, such mechanism reversely strengthens the weightiness of CSC targeting in EC administration with individualized therapy.

Eventually, emphasizing the importance of estimating prospective therapy response in precision medicine is warranted. For instance, a better OS in recurrent glioblastoma patients than previous data was achieved by using a systemic assay which firstly stratifies the cell kill rate to find an efficient one among different drug combinations targeting both CSCs and non-CSC cells [215]. This frontier attempt provided an auspicious way for the future treatment of recurrent EC patients or those with poor estimated prognosis.

## 6. Conclusions and Future Perspectives

The focus on ECSCs opens a new vision for translational research of EC and may result in further understanding of key mechanisms of EC etiology, progression, recurrence, and therapy resistance. As we discussed, CSCs could be isolated from esophageal tumors using either ECSC biomarker based or ECSC biomarker-free methods. A combined application of multiple markers or multiple methods to screen for CSCs in future studies may help to overcome the limitations derived from the heterogeneity of individual tumors. Furthermore, we emphasize the importance and advantages of integration of single cell analysis in esophageal cancer stem cell studies. This new method will help to understand both intertumor and intratumor heterogeneity of CSCs in EC and the clonal architecture of esophageal cancer for both adeno- and squamous cell carcinomas. The crosstalk between ECSCs and their niche (TME) not only plays a pivotal role during oncogenesis but also has profound effects on modulating therapeutic efficacy. Therefore, future strategies of combined treatments, that target CSCs and the TME may result in successful implementation of individualized therapy of EC patients.

## Figures and Tables

**Figure 1 cells-09-01481-f001:**
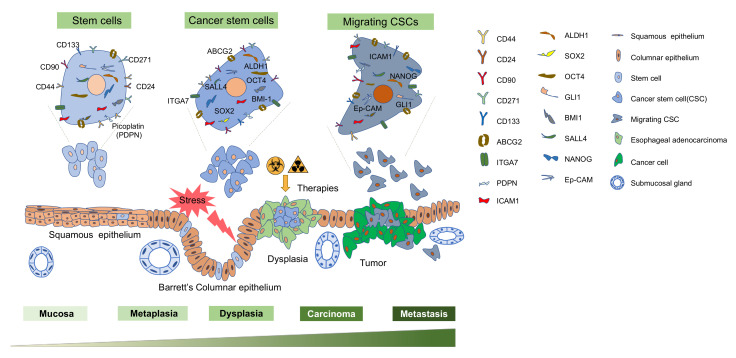
A schematic of various esophageal cancer stem cell markers and the overview of the progression from normal squamous epithelium to dysplastic cell and finally developing into adenocarcinoma. Cell surface markers CD44, CD24, CD90, CD271, CD133, ABCG2, ITGA7, ICAM-1 and PDPN are used as single markers while CD44 and CD133 can be used in combination with other markers, including CD24, CD133 and CXCR4 to identify cancer stem cells (CSCs). Others are the transcription factors BMI1, NANOG, SOX2, OCT-4, GLI-1, SALL4, and Ep-CAM are implicated to enrich the CSCs. Various cell types have been proposed to give rise to metaplasia (the replacement of esophageal squamous epithelium by Barrett’s columnar epithelium in response to esophageal injury) which can progress to esophageal adenocarcinoma. Esophageal cancer cells are heterogeneous and include cancer stem cell populations. Chemotherapy and/or radiotherapy kill differentiated cancer cells but may fail to kill CSCs, which arise from stem cells, progenitor cells, or differentiated cells. Migrating cancer stem cells are considered to have a crucial role in initiating cancer metastasis.

**Table 1 cells-09-01481-t001:** Cancer stem cell markers for prognosis of esophageal cancer.

Markers	Cancer Type EAC/ESCC	Results	Marker for Diagnosis or Prognosis	Reference
Single marker				
CD44	EAC ESCC	Cell surface protein: contributes to tumor invasion and regulates EMT	High CD44 expression correlates to positive lymph node ratio and lymph vascular invasion	[29,30,31]
ABCG2	ESCC	ATP-binding cassette transporter (membrane transporter) is associated with the drug resistance and metastasis	The presence of ABCG2-positive cells was associated with poor survival independent of primary tumor size and positive lymph node metastasis	[33,46,47]
ALDH1	EAC ESCC	Intracellular enzyme oxidizing aldehydes: ALDH1^+^ cancer cells possess highly invasive and metastatic capabilities with EMT phenotype and are associated with therapy resistances	Positive ALDH1 staining was relevant to higher clinical stage and shorter survival time	[43,44,45]
CD133	ESCC	Cell surface protein: promotes tumor initiation and self-renewal capacity as well as chemoresistance.	The presence of CD133^+^ cancer cells was associated with tumor cell differentiation	[32,33,34]
CD271	ESCC	Cell surface protein: CD271^+^ cancer cells possess higher self-renewal activity and are associated with therapy-resistance and lymphnode metastasis	Ep-CAM+ CD271(p75NTR)+ tumor cells in peripheral blood correlate with clinically diagnosed metastasis and venous invasion	[35,36,37]
LgR5	EAC	Cell surface protein: promotes proliferation, migration and invasion ability	High LgR5 was associated with worse survival	[38,39,40]
CD90	ESCC	Cell surface protein: CD90^+^ cells possess higher self-renewal activity and metastatic potential, and are more resistant to chemotherapy	Higher CD90 expression exhibit more local invasion and distant metastasis, indicating a poor prognosis	[41,42]
ITGA7	ESCC	Cell surface receptor: ITGA7 contributes to tumor innitiation and drug resistance, it promotes metastasis via inducing EMT together with an anti-apoptosis function.	More ITGA7^+^ cells in ESCC tissues predict a worse prognosis	[49]
ICAM1	ESCC	Intercellular adhesion molecule1: promotes cancer cell migration, invasion, EMT, sphere formation, tumorigenesis and drug resistance		[48]
SOX2	ESCC	Transcription factor: promotes cancer cells migration and invasion as well as chemoresistance to cisplatin	Controversial results exist regarding the prognostic value of SOX2 because of opposite conclusion among studies	[53,54,55,56,57]
NANOG	ESCC	Transcriptional regulator: regulates cancer cells proliferation and drug resistance		[58,59,60]
BMI-1	ESCC	Transcriptional regulator: regulates radiosensitivity of tumor cells and inbitits cell growth and invasion	Overexpression of BMI-1 is associated with progression and invasion of EC	[61,62,63,64]
OCT-4	ESCC	Transcriptional regulator: promotes cell cycle progression and accelerates proliferation and invasion of esophageal cancer cells	Overexpression of OCT-4 is significantly associated with higher histological grade and poorer survival	[54,62,65,66]
Ep-CAM	ESCC	Transmembrane glycoprotein: Ep-CAM contributes to cell proliferation and tumorigenesis	Expression level of Ep-CAM inversely correlates with degree of differentiation	[67,68,69]
Gli-1	EAC ESCC	Transcription factor: promotes cell proliferation and is associated with chemoradiation resistance	Gli-1 is positively associated with distant metastasis, indicates poor outcome	[70,71,72]
SALL4	ESCC	Transcription factor: promotes cell proliferation, migration and invasion as well as chemoresistance to cisplatin, contributes to tumorigenesis in vivo	Overexpression of SALL4 was found in a majority of ESCC tissues and correlates with poor survival	[55,73]
Podoplanin (PDPN)	ESCC	Transmembrane protein: accelerates the proliferation and regulates tumor EMT	PDPN expression at the edge of cancer cell nest associates with tumor invasion and poor prognosis	[74,75,76,77]
Combined markers				
CD44^+^/CD24^−^	EAC and ESCC	CD44^+^/CD24^−^ EC cells exert a higher proliferation rate and mediate therapy resistance		[50]
CD44^+^/CD133^+^	ESCC		Strong expression of CD44 and CD133 indicates a poor prognosis regardless of chemotherapy in ESCC	[51]
CD133^+^/CXCR4^+^	ESCC	CD133^+^CXCR4^+^cells regulate tumor invasion and show high proliferative capacity	Concomitant CD133-CXCR4 expression heralds impaired disease-free survival and overall survival	[52]

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
