# Peer review of "Linking Cancer Stem Cell Plasticity to Therapeutic Resistance-Mechanism and Novel Therapeutic Strategies in Esophageal Cancer"

_cells, 2020, doi:10.3390/cells9061481_

Round 1
Reviewer 1 Report
Zhou et al. reviewed current research and knowledge on the cancer stem cells (CSCs) in esophageal cancer and their potential role as therapeutic targets. The topic of the review is interesting and timely; however, additional details and editing would improve the manuscript.
The review lacks details on how the authors planned the review of the literature, what information was included or excluded, whether levels of evidence were used in assessing the value of each publication selected for inclusion, and whether unpublished material was included.
The authors give a very superficial overview of CSC analysis mainly focusing on in vitro assays and single cell analysis whereas the gold standard CSC in vivo assays such as tumor transplantation, limiting dilution analysis and lineage-tracing assay are neglected and should be discussed.
Authors are also encouraged to mention that sphere forming capacity is not always indicative for CSC enrichment, and the same is true for acquired therapy resistance.
An overview table for the esophageal CSC markers with their projection in the clinic field as potential prognostic markers would be appreciated.
References are not limited to recent works (> 40% published before 2015) and the cited literature is mainly focused on other tumor types, but not esophageal cancers. Some critical manuscripts e.g. Wang D et al. & Coppes RP. Cancers, 2019; Zhang HF et al. Stem Cells. 2016 are not mentioned.
A commentary should be added on the possible role of HPV infection for the activation of self-renewal pathways and CSC expansion.
Author Response
To reviewer 1#:
Dear Reviewer,
We appreciated all your input, which was of the greatest value to our manuscript, it has led us to better understand the strength and weaknesses of our manuscript. We have made our best to answer your comments and made appropriate changes according to your guidance.
Below are our response and updates for your review:
Comment 1: The review lacks details on how the authors planned the review of the literature, what information was included or excluded, whether levels of evidence were used in assessing the value of each publication selected for inclusion, and whether unpublished material was included.
Answer 1: Thank you for the reminder. Actually, it is crucial for a review article to give the details about the analytic strategy of the relevant literature, which can provide a full view of this review and also make it easier to understand the concept of this article. We apologized that we ignored this point before. We now add the specification of literature selection in the revision.
Changes 1: In the Introduction section, a brief description about how we select the literature was added in the manuscript (Page 2, line 74-78).
Comment 2:
The authors give a very superficial overview of CSC analysis mainly focusing on in vitro assays and single cell analysis whereas the gold standard CSC in vivo assays such as tumor transplantation, limiting dilution analysis and lineage-tracing assay are neglected and should be discussed.
Answer 2: We apologized for the limits in our CSC analysis part which was mainly focus on in vitro assays. We agree that it is more convincing to apply in vivo assays regarding CSC studies in this review. Thus, we have discussed the application of tumor transplantation, limiting dilution analysis and lineage-tracing assay on esophageal cancer stem cells (if available) to improve the manuscript.
Changes 2: In the ECSC isolation - biomarker-free section, we add new context and the citation about tumor transplantation with limiting dilution analysis and lineage-tracing assay on esophageal cancer studies (Page 6-7, line 153-167).
Comment 3: Authors are also encouraged to mention that sphere forming capacity is not always indicative for CSC enrichment, and the same is true for acquired therapy resistance.
Answer 3: We sincerely appreciated for the precious advice, some studies have demonstrated that the spheres even did not consist of CSCs, and the CSC enrichment could be achieved by acquired therapy resistance. For example: Two apparently distinct populations might be the cells in different proliferating status or cell cycles. And this assay also has some other limitations, so we also mention the shortcoming of this method in our revised manuscript.
Changes 3: In the ECSC isolation - biomarker-free section, a sentence was added with the related reference: “Eyes Wide Open: A Critical Review of Sphere-Formation as an Assay for Stem Cells” (Page 6, line 1400-142).
Comment 4: An overview table for the esophageal CSC markers with their projection in the clinic field as potential prognostic markers would be appreciated.
Answer 4: It is truly a thoughtful suggestion. Increasing experimental and analytical evidence are declared in past decades to support a bright prospect of the CSC markers as potential diagnostic or prognostic markers for clinical purpose. CSC markers are becoming promising tools for both translational researches to understand the tumor biology and clinical applications to benefit cancer patients.
Changes 4: In the Table 1, a separate column was added to summarize the diagnostic or prognostic values of some popular CSC markers based on recent papers (Page 3-6, line 112).
Comment 5: References are not limited to recent works (> 40% published before 2015) and the cited literature is mainly focused on other tumor types, but not esophageal cancers. Some critical manuscripts e.g. Wang D et al. & Coppes RP. Cancers, 2019; Zhang HF et al. Stem Cells. 2016 are not mentioned.
Answer 5: We appreciated for the valuable advice. In order to better understand its latest research results and developments, we should indeed cite latest references for research and analysis. Considering that the overall research on esophageal cancer is quite insufficient, so we try our best to delete some references before the year of 2015 and add some new publications including Wang D et al. & Coppes RP. Cancers, 2019; Zhang HF et al. Stem Cells. 2016 in our review according to the reviewer’s comments.Additionally, we have added more details about the Hh and PI3K/AKT/c-MYC signaling pathway in our review to increase the solidity of the manuscript.
Changes 5: In the ECSC signaling pathways section, some details about these two pathways on esophageal cancer stem cell are described from two critical manuscripts about Hh and PI3K/AKT/c-MYC signaling pathway. We make the change in Page 7-9, line 201, 232-237, 246, 252-262.
Comment 6: A commentary should be added on the possible role of HPV infection for the activation of self-renewal pathways and CSC expansion.
Answer 6: Thanks to the reviewer’s suggestions. HPV is an important etiologic factor associated with the development of several types of cancers. However, in esophageal cancer, the etiology of HPV infection and tumorigenesis is still controversial. Considering our manuscript mainly focus on the CSCs in esophageal cancer, after deliberations among the authors, we decided to add a brief hint regarding HPV infection in our manuscript and we hope the reviewer find it appropriate. We appreciated the inspiration from the reviewer of linking the virus infection and CSC regulations.
Changes 6: In the ECSC signaling pathways section, a brief description towards HPV and CSCs was added in the manuscript (Page 8-9, line 256-262).
In all, we would like to thank the reviewer for the patience and concern, which helped to improve our manuscript!
Best Regards,
Yue Zhao
Department of General, Visceral and Cancer Surgery, University Hospital of Cologne, Kerpener Straße 62, 50937, Cologne, Germany.
Reviewer 2 Report
Comments on the manuscript “Linking Cancer Stem Cell Plasticity to Therapeutic Resistance-
Mechanism and Novel Therapeutic Strategies in Esophageal Cancer”
In this review, the authors have performed a review to evaluate current evidence on CSCs in esophageal cancer.
The main problem remains the tumor response on multimodality treatment with even a considerable rate of recurrent disease in ypCR (ypT0N0) cases.
Based on the literature the authors should explain the development of recurrences and the role of cancer stem cells (CSC) with the complete response of cancer cells and repopulation of CSC. The role of CSC in these cases is still not clear and the authors should give us more insight in their mechanism and the future strategy to treat esophageal cancer taken into account specific characteristics of CSCs.
Moreover, some information in selecting the literature will give the reader a better view of the scientific level of these papers.
Author Response
Dear Reviewer,
We appreciate your kind words and constructive comments, please find below the detailed response to each comment and the respective revision.
Comment 1&2: Based on the literature the authors should explain the development of recurrences and the role of cancer stem cells (CSC) with the complete response of cancer cells and repopulation of CSC. The role of CSC in these cases is still not clear and the authors should give us more insight in their mechanism and the future strategy to treat esophageal cancer taken into account specific characteristics of CSCs.
Answer 1&2: Thank you for the very important comment. It is mentioned by dozens of reviews that the CSCs should partially take the responsibility for cancer recurrence in different tumor types. Recently, some groups presented direct evidence indicating the tumor initiating property of stem cell like cells on cancer recurrence from different solid tumors. Indeed, increasing numbers of clinical trials and preclinical works have been conducted or are undergoing to target CSCs alone or in combination with conventional therapies or other target therapies. Some encouraging output are obtained while more trials are still waiting to be explored in esophageal cancer. Concerning the role of CSCs in conventional chemotherapeutic or chemoradiation resistance, we updated the part 5. Therapeutic strategies targeting CSC in EC to present recent effort mainly on EC as well as on other cancer types.
Changes 1&2: Direct evidences were added based on two research works to support the causal link of CSC and tumor recurrence (Page11, line393-401). Concerning the rising demand of personalized medicines for EC administration nowadays, we additionally remind the reader by the end of section 5. Therapeutic strategies targeting CSC in EC that combined therapy targeting CSCs might obtain a better response and further prevent the recurrence (Page11, line 402-406).
Comment 3: Moreover, some information in selecting the literature will give the reader a better view of the scientific level of these papers.
Answer 3: Thank you for the nice advice. We have added the strategy of literature selection for inclusion in the review.
Changes 3: In the Introduction section, a brief description about how we select the literature was added in the manuscript (Page 2, line 74-78).
Again, we would like to express our great gratitude to the reviewer for taking the time and effort to work on our manuscript and provide constructive advice to improve this paperwork.
With our kindest regards,
Yue Zhao
Department of General, Visceral and Cancer Surgery, University Hospital of Cologne, Kerpener Straße 62, 50937, Cologne, Germany.
Round 2
Reviewer 1 Report
The authors have addressed all my concerns, and I believe that the manuscript is ready to be accepted for publication in Cells journal.